# PARROT: DATA-DRIVEN BEHAVIORAL PRIORS FOR REINFORCEMENT LEARNING

**Avi Singh**[*], **Huihan Liu**[*], **Gaoyue Zhou, Albert Yu, Nicholas Rhinehart, Sergey Levine**
University of California, Berkeley

## ABSTRACT

Reinforcement learning provides a general framework for flexible decision making and control, but requires extensive data collection for each new task that an agent needs to learn. In other machine learning fields, such as natural language processing or computer vision, pre-training on large, previously collected datasets to bootstrap learning for new tasks has emerged as a powerful paradigm to reduce data requirements when learning a new task. In this paper, we ask the following question: how can we enable similarly useful pre-training for RL agents? We propose a method for pre-training behavioral priors that can capture complex input-output relationships observed in successful trials from a wide range of previously seen tasks, and we show how this learned prior can be used for rapidly learning new tasks without impeding the RL agent's ability to try out novel behaviors. We demonstrate the effectiveness of our approach in challenging robotic manipulation domains involving image observations and sparse reward functions, where our method outperforms prior works by a substantial margin. Additional materials can be found on our project website: https://sites.google.com/view/parrot-rl

## 1 INTRODUCTION

Reinforcement Learning (RL) is an attractive paradigm for robotic learning because of its flexibility in being able to learn a diverse range of skills and its capacity to continuously improve. However, RL algorithms typically require a large amount of data to solve each individual task, including simple ones. Since an RL agent is generally initialized without any prior knowledge, it must try many largely unproductive behaviors before it discovers a high-reward outcome. In contrast, humans rarely attempt to solve new tasks in this way: they draw on their prior experience of what is *useful* when they attempt a new task, which substantially shrinks the task search space. For example, faced with a new task involving objects on a table, a person might grasp an object, stack multiple objects, or explore other object rearrangements, rather than re-learning how to move their arms and fingers.

Can we endow RL agents with a similar sort of *behavioral prior* from past experience? In other fields of machine learning, the use of large prior datasets to bootstrap acquisition of new capabilities has been studied extensively to good effect. For example, language models trained on large, diverse datasets offer representations that drastically improve the efficiency of learning downstream tasks (Devlin et al., 2019). What would be the analogue of this kind of pre-training in robotics and RL? One way we can approach this problem is to leverage successful trials from a wide range of previously seen tasks to improve learning for *new* tasks. The data could come from previously learned policies, from human demonstrations, or even unstructured teleoperation of robots (Lynch et al., 2019). In this paper, we show that behavioral priors can be obtained through *representation learning*, and the representation in question must not only be a representation of inputs, but actually a representation of input-output relationships – a space of possible and likely mappings from states to actions among which the learning process can interpolate when confronted with a new task.

What makes for a good representation for RL? Given a new task, a good representation must (a) provide an effective exploration strategy, (b) simplify the policy learning problem for the RL algorithm, and (c) allow the RL agent to retain full control over the environment. In this paper, we address

---

[*]Equal contribution. Correspondence to Avi Singh (avisingh@berkeley.edu).

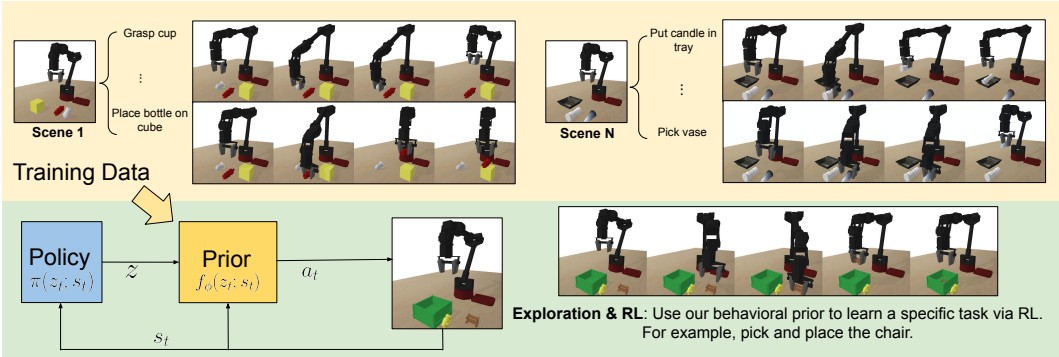

Figure 1: **Our problem setting**. Our training dataset consists of near-optimal state-action trajectories (without reward labels) from a wide range of tasks. Each task might involve interacting with a different set of objects. Even for the same set of objects, the task can be different depending on our objective. For example, in the upper right corner, the objective could be picking up a cup, or it could be to place the bottle on the yellow cube. We learn a behavioral prior from this multi-task dataset capable of trying many different useful behaviors when placed in a new environment, and can aid an RL agent to quickly learn a specific task in this new environment.

all of these challenges through learning an *invertible* function that maps noise vectors to complex, high-dimensional environment actions. Building on prior work in normalizing flows (Dinh et al., 2017), we train this mapping to maximize the (conditional) log-likelihood of actions observed in successful trials from past tasks. When dropped into a new MDP, the RL agent can now sample from a unit Gaussian, and use the learned mapping (which we refer to as the *behavioral prior*) to generate likely environment actions, conditional on the current observation. This learned mapping essentially transforms the original MDP into a simpler one for the RL agent, as long as the original MDP shares (partial) structure with previously seen MDPs (see Section 3). Furthermore, since this mapping is invertible, the RL agent still retains full control over the original MDP: for every possible environment action, there exists a point within the support of the Gaussian distribution that maps to that action. This allows the RL agent to still try out new behaviors that are distinct from what was previously observed.

Our main contribution is a framework for pre-training in RL from a diverse multi-task dataset, which produces a behavioral prior that accelerates acquisition of new skills. We present an instantiation of this framework in robotic manipulation, where we utilize manipulation data from a diverse range of prior tasks to train our behavioral prior, and then use it to bootstrap exploration for new tasks. By making it possible to pre-train action representations on large prior datasets for robotics and RL, we hope that our method provides a path toward leveraging large datasets in the RL and robotics settings, much like language models can leverage large text corpora in NLP and unsupervised pre-training can leverage large image datasets in computer vision. Our method, which we call Prior AcceleRated ReinfOrcemenT (PARROT), is able to quickly learn tasks that involve manipulating previously unseen objects, from image observations and sparse rewards, in settings where RL from scratch fails to learn a policy at all. We also compare against prior works that incorporate prior data for RL, and show that PARROT substantially outperforms these prior works.

## 2 RELATED WORK

**Combining RL with demonstrations.** Our work is related to methods for learning from demonstrations (Pomerleau, 1989; Schaal et al., 2003; Ratliff et al., 2007; Pastor et al., 2009; Ho & Ermon, 2016; Finn et al., 2017b; Giusti et al., 2016; Sun et al., 2017; Zhang et al., 2017; Lynch et al., 2019). While demonstrations can also be used to speed up RL (Schaal, 1996; Peters & Schaal, 2006; Kormushev et al., 2010; Hester et al., 2017; Vecerík et al., 2017; Nair et al., 2018; Rajeswaran et al., 2018; Silver et al., 2018; Peng et al., 2018; Johannink et al., 2019; Gupta et al., 2019), this usually requires collecting demonstrations for the specific task that is being learned. In contrast, we use data from a wide range of *other* prior tasks to speed up RL for a *new* task. As we show in our experiments, PARROT is better suited to this problem setting when compared to prior methods that combine imitation and RL for the same task.

**Generative modeling and RL.** Several prior works model multi-modal action distributions using expert trajectories from different tasks. IntentionGAN (Hausman et al.) and InfoGAIL (Li et al., 2017) learn multi-modal policies via interaction with an environment using an adversarial imitation approach (Ho & Ermon, 2016), but we learn these distributions only from data. Other works learn these distributions from data (Xie et al., 2019; Rhinehart et al., 2020) and utilize them for planning at test time to optimize a user-provided cost function. In contrast, we use the behavioral prior to augment model-free RL of a new task. This allows us to learn policies for new tasks that may be substantially different from prior tasks, since we can collect data specific to the new task at hand, and we do not explicitly need to model the environment, which can be complicated for high-dimensional state and action spaces, such as when performing continuous control from images observations. Another line of work (Ghadirzadeh et al., 2017; Hämäläinen et al., 2019; Ghadirzadeh et al., 2020) explores using generative models for RL, using a variational autoencoder (Kingma & Welling, 2014) to model entire trajectories in an observation-independent manner, and then learning an open-loop, single-step policy using RL to solve the downstream task. Our approach differs in several key aspects: (1) our model is observation-conditioned, allowing it to prioritize actions that are relevant to the current scene or environment, (2) our model allows for closed-loop feedback control, and (3) our model is *invertible*, allowing the high-level policy to retain full control over the action space. Our experiments demonstrate these aspects are crucial for solving harder tasks.

**Hierarchical learning.** Our method can be interpreted as training a hierarchical model: the low-level policy is the behavioral prior trained on prior data, while the high-level policy is trained using RL and controls the low-level policy. This structure is similar to prior work in hierarchical RL (Dayan & Hinton, 1992; Parr & Russell, 1997; Dietterich, 1998; Sutton et al., 1999; Kulkarni et al., 2016). We divide prior work in hierarchical learning into two categories: methods that seek to learn both the low-level and high-level policies through active interaction with an environment (Kupcsik et al., 2013; Heess et al., 2016; Bacon et al., 2017; Florensa et al., 2017; Haarnoja et al., 2018a; Nachum et al., 2018; Chandak et al., 2019; Peng et al., 2019), and methods that learn temporally extended actions, also known as *options*, from demonstrations, and then recompose them to perform long-horizon tasks through RL or planning (Fox et al., 2017; Krishnan et al., 2017; Kipf et al., 2019; Shankar et al., 2020; Shankar & Gupta, 2020). Our work shares similarities with the data-driven approach of the latter methods, but work on options focuses on modeling the *temporal* structure in demonstrations for a small number of long-horizon tasks, while our behavioral prior is not concerned with temporally-extended abstractions, but rather with transforming the original MDP into one where potentially useful behaviors are more likely, and useless behaviors are less likely.

**Meta-learning.** Our goal in this paper is to utilize data from previously seen tasks to speed up RL for new tasks. Meta-RL (Duan et al., 2016; Wang et al., 2016; Finn et al., 2017a; Mishra et al., 2017; Rakelly et al., 2019; Mendonca et al., 2019; Zintgraf et al., 2020; Fakoor et al., 2020) and meta-imitation methods (Duan et al., 2017; Finn et al., 2017c; Huang et al., 2018; James et al., 2018; Paine et al., 2018; Yu et al., 2018; Huang et al., 2019; Zhou et al., 2020) also seek to speed up learning for new tasks by leveraging experience from previously seen tasks. While meta-learning provides an appealing and principled framework to accelerate acquisition of future tasks, we focus on a more lightweight approach with relaxed assumptions that make our method more practically applicable, and we discuss these assumptions in detail in the next section.

## 3 PROBLEM SETUP

Our goal is to improve an agent's ability to learn new tasks by incorporating a behavioral prior, which it can acquire from previously seen tasks. Each task can be considered a Markov decision process (MDP), which is defined by a tuple $(\mathcal{S}, \mathcal{A}, \mathrm{T}, r, \gamma)$, where $\mathcal{S}$ and $\mathcal{A}$ represent state and action spaces, $\mathrm{T}(s'|s, a)$ and $r(s, a)$ represent the dynamics and reward functions, and $\gamma \in (0, 1)$ represents the discount factor. Let $p(M)$ denote a distribution over such MDPs, with the constraint that the state and action spaces are fixed. In our experiments, we treat high-dimensional images as $s$, which means that this constraint is not very restrictive in practice. In order for the behavioral prior to be able to accelerate the acquisition of new skills, we assume the behavioral prior is trained on data that structurally resembles potential optimal policies for all or part of the new task. For example, if the new task requires placing a bottle in a tray, the prior data might include some behaviors that involve picking up objects. There are many ways to formalize this assumption. One way to state this formally is to assume that prior data consists of executions of near-optimal policies for MDPs

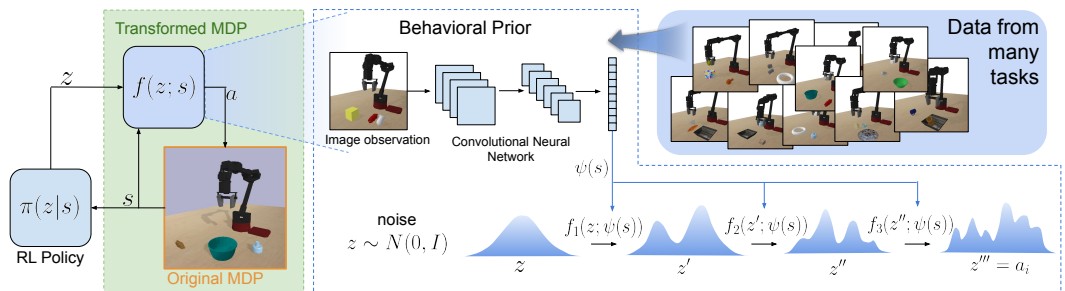

**Figure 2: PARROT.** Using successful trials from a large variety of tasks, we learn an invertible mapping $f_\phi$ that maps noise $z$ to useful actions $a$. This mapping is conditioned on the current observation, which in our case is an RGB image. The image is passed through a stack of convolutional layers and flattened to obtain an image encoding $\psi(s)$, and this image encoding is then used to condition each individual transformation $f_i$ of our overall mapping function $f_\phi$. The parameters of the mapping (including the convolutional encoder) are learned through maximizing the conditional log-likelihood of state-action pairs observed in the dataset. When learning a new task, this mapping can simplify the MDP for an RL agent by mapping actions sampled from a randomly initialized policy to actions that are likely to lead to useful behavior in the current scene. Since the mapping is invertible, the RL agent still retains full control over the action space of the original MDP, simply the likelihood of executing a useful action is increased through use of the pre-trained mapping.

drawn according to $M \sim p(M)$, and the new task $M^\star$ is likewise drawn from $p(M)$. In this case, the generative process for the prior data can be expressed as:

$$M \sim p(M), \quad \pi_M(\tau) = \arg\max_\pi \mathbb{E}_{\pi,M}[R_M], \quad \tau_M \sim \pi_M(\tau), \tag{1}$$

where $\tau_M = (s_1, a_1, s_2, a_2, \ldots, s_T, a_T)$ is a sequence of state and actions, $\pi_M(\tau)$ denotes a near-optimal policy (Kearns & Singh, 2002) for MDP $M$ and $R_M = \sum_{t=0}^\infty \gamma^t r_t$. When incorporating the behavioral prior for learning a new task $M^\star$, our goal is the same as standard RL: to find a policy $\pi$ that maximizes the expected return $\arg\max_\pi \mathbb{E}_{\pi,M^\star}[R_{M^\star}]$. Our assumption on tasks being drawn from a distribution $p(M)$ shares similarities with the meta-RL problem (Wang et al., 2016; Duan et al., 2016), but our setup is different: it does not require accessing any task in $p(M)$ except the new task we are learning, $M^\star$. Meta-RL methods need to interact with the tasks in $p(M)$ during meta-training, with access to rewards and additional samples, whereas we learn our behavioral prior simply from data, without even requiring this data to be labeled with rewards. This is of particular importance for real-world problem settings such as robotics: it is much easier to store data from prior tasks (e.g., different environments) than to have a robot physically revisit those prior settings and retry those tasks, and not requiring known rewards makes it possible to use data from a variety of sources, including human-provided demonstrations. In our setting, RL is performed in only one environment, while the prior data can come from many environments.

Our setting is related to meta-imitation learning (Duan et al., 2017; Finn et al., 2017c), as we speed up learning new tasks using data collected from past tasks. However, meta-imitation learning methods require at least one demonstration for each new task, whereas our method can learn new tasks without any demonstrations. Further, our data requirements are less stringent: meta-imitation learning methods require all demonstrations to be optimal, require all trajectories in the dataset to have a task label, and requires "paired demonstrations", i.e. at least two demonstrations for each task (since meta-imitation methods maximize the likelihood of actions from one demonstration after conditioning the policy on another demonstration from the same task). Relaxing these requirements increases the scalability of our method: we can incorporate data from a wider range of sources, and we do not need to explicitly organize it into specific tasks.

## 4 BEHAVIORAL PRIORS FOR REINFORCEMENT LEARNING

Our method learns a behavioral prior for downstream RL by utilizing a dataset $\mathcal{D}$ of (near-optimal) state-action pairs from previously seen tasks. We do so by learning a state-conditioned mapping $f_\phi : \mathcal{Z} \times \mathcal{S} \rightarrow \mathcal{A}$ (where $\phi$ denotes learnable parameters) that transforms a noise vector $z$ into an action $a$ that is likely to be useful in the current state $s$. This removes the need for exploring via "meaningless" random behavior, and instead enables an exploration process where the agent attempts behaviors that have been shown to be useful in previously seen domains. For example, if a

robotic arm is placed in front of several objects, randomly sampling $z$ (from a simple distribution, such as the unit Gaussian) and applying the mapping $a = f_\phi(z; s)$ should result in actions that, when executed, result in meaningful interactions with the objects. This learned mapping essentially transforms the MDP experienced by the RL agent into a simpler one, where every random action executed in this transformed MDP is much more likely to lead to a useful behavior.

How can we learn such a mapping? In this paper, we propose to learn this mapping through state-conditioned generative modeling of the actions observed in the original dataset $\mathcal{D}$, and we refer to this state-conditioned distribution over actions as the behavioral prior $p_{\text{prior}}(a|s)$. A deep generative model takes noise as input, and outputs a plausible sample from the target distribution, i.e. it can represent $p_{\text{prior}}(a|s)$ as a distribution over noise $z$ using a deterministic mapping $f_\phi : \mathcal{Z} \times \mathcal{S} \mapsto A$ When learning a new task, we can use this mapping to reparametrize the action space of the RL agent: if the action chosen by the randomly initialized neural network policy is $z$, then we execute the action $a = f_\phi(z; s)$ in the original MDP, and learn a policy $\pi(z|s)$ that maximizes the task reward through learning to control the inputs to the mapping $f_\phi$. The training of the behavioral prior and the task-specific policy is decoupled, allowing us to mix and match RL algorithms and generative models to best suit the application of interest. An overview of our overall architecture is depicted in Figure 2. In the next subsection, we discuss what properties we would like the behavioral prior to satisfy, and present one particular choice for learning a prior that satisfies all of these properties.

## 4.1 LEARNING A BEHAVIORAL PRIOR WITH NORMALIZING FLOWS

For the behavioral prior to be effective, it needs to satisfy certain properties. Since we learn the prior from a *multi-task* dataset, containing several different behaviors even for the same initial state, the learned prior should be capable of representing complex, multi-modal distributions. Second, it should provide a mapping for generating "useful" actions from noise samples when learning a new task. Third, the prior should be state-conditioned, so that only actions that are relevant to the current state are sampled. And finally, the learned mapping should allow easier learning in the reparameterized action space *without* hindering the RL agent's ability to attempt novel behaviors, including actions that might not have been observed in the dataset $\mathcal{D}$. Generative models based on normalizing flows (Dinh et al., 2017) satisfy all of these properties well: they allow maximizing the model's exact log-likelihood of observed examples, and learn a deterministic, invertible mapping that transforms samples from a simple distribution $p_z$ to examples observed in the training dataset. In particular, the real-valued non-volume preserving (real NVP) architecture introduced by Dinh et al. (2017) allows using deep neural networks to parameterize this mapping (making it expressive) While the original real NVP work modelled unconditional distributions, follow-up work has found that it can be easily extended to incorporate conditioning information (Ardizzone et al., 2019). We refer the reader to prior work (Dinh et al., 2017) for a complete description of real NVPs, and summarize its key features here. Given an invertible mapping $a = f_\phi(z; s)$, the change of variable formula allows expressing the likelihood of the observed actions using samples from $\mathcal{D}$ in the following way:

$$p_{\text{prior}}(a|s) = p_z\left(f_\phi^{-1}(a; s)\right) \left|\det\left(\partial f_\phi^{-1}(a; s)/\partial a\right)\right| \qquad (2)$$

Dinh et al. (2017) propose a particular (unconditioned) form of the invertible mapping $f_\phi$, called an affine coupling layer, that maintains tractability of the likelihood term above, while still allowing the mapping $f_\phi$ to be expressive. Several coupling layers can be composed together to transform simple noise vectors into samples from complex distributions, and each layer can be conditioned on other variables, as shown in Figure 2.

## 4.2 ACCELERATED REINFORCEMENT LEARNING VIA BEHAVIORAL PRIORS

After we obtain the mapping $f_\phi(z; s)$ from the behavioral prior learned by maximizing the likelihood term in Equation 2, we would like to use it to accelerate RL when solving a new task. Instead of learning a policy $\pi_\theta(a|s)$ that directly executes its actions in the original MDP, we learn a policy $\pi_\theta(z|s)$, and execute an action in the environment according to $a = f_\phi(z; s)$. As shown in Figure 2, this essentially transforms the MDP experienced for the RL agent into one where random actions $z \sim p_z$ (where $p_z$ is the base distribution used for training the mapping $f_\phi$) are much more likely to result in useful behaviors. To enable effective exploration at the start of the learning period, we initialize the RL policy to the base distribution used for training the prior, so that at the beginning of training, $\pi_\theta(z|s) := p_z(z)$. Since the mapping $f_\phi$ is invertible, the RL agent still retains full control

over the action space: for any given $a$, it can always find a $z$ that generates $z = f_\phi^{-1}(a; s)$ in the original MDP. The learned mapping increases the likelihood of useful actions without crippling the RL agent, making it ideal for fine-tuning from task-specific data. Our complete method is described in Algorithm 1 in Appendix A. Note that we need to learn the mapping $f_\phi$ only once, and it can be used for accelerated learning of any new task.

## 4.3 IMPLEMENTATION DETAILS

We use real NVP to learn $f_\phi$, and as shown in Figure 2, each coupling layer in the real NVP takes as input the the output of the previous coupling layer, and the conditioning information. The conditioning information in our case corresponds to RGB image observations, which allows us to train a single behavioral prior across a wide variety of tasks, even when the tasks might have different underlying states (for example, different objects). We train a real NVP model with four coupling layers; the exact architecture, and other hyperparameters, are detailed in Appendix B. The behavioral prior can be combined with any RL algorithm that is suitable for continuous action spaces, and we chose to use the soft actor-critic (Haarnoja et al., 2018b) due to its stability and ease of use.

## 5 EXPERIMENTS

Our experiments seek to answer: **(1)** Can the behavioral prior accelerate learning of *new* tasks? **(2)** How does PARROT compare to prior works that accelerate RL with demonstrations? **(3)** How does PARROT compare to prior methods that combine hierarchical imitation with RL?

**Domains.** We evaluate our method on a suite of challenging robotic manipulation tasks, a subset of which are depicted in Figure 3. Each task involves controlling a 6-DoF robotic arm and its gripper, with a 7D action space. The observation is a $48 \times 48$ RGB image, which allows us to use the same observation representation across tasks, even though each underlying task might have a different underlying state (e.g., different objects). No other observations (such as joint angles or end-effector positions) are provided. In each task, the robot needs to interact with one or two objects in the scene to achieve its objective, and there are three objects in each scene. Note that all of the objects in the test scenes are *novel* – the dataset $\mathcal{D}$ contains no interactions with these objects. The object positions at the start of each trial are randomized, and the policy must infer these positions from image observations in order to successfully solve the task. A reward of $+1$ is provided when the objective for the task is achieved, and the reward is zero otherwise. Detailed information on the objective for each task and example rollouts are provided in Appendix C.1, and on our anonymous project website[1].

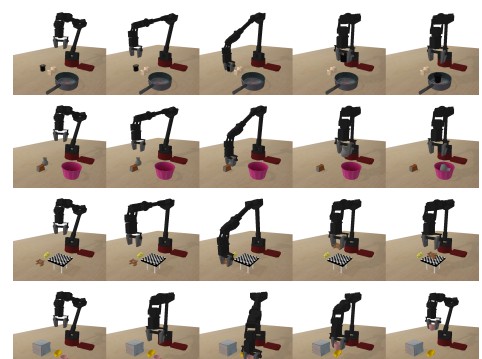

Figure 3: **Tasks.** A subset of our evaluation tasks, with one task shown in each row. In the first task (first row), the objective is to pick up a can and place it in the pan. In the second task, the robot must pick up the vase and put it in the basket. In the third task, the goal is to place the chair on top of the checkerboard. In the fourth task, the robot must pick up the mug and hold it above a certain height. Initial positions of all objects are randomized, and must be inferred from visual observations. Not all objects in the scene are relevant to the current task.

**Data collection.** Our behavioral prior is trained on a diverse dataset of trajectories from a wide range of tasks, and then utilized to accelerate reinforcement learning of new tasks. As discussed in Section 3, for the prior to be effective, it needs to be trained on a dataset that structurally resembles the behaviors that might be optimal for the new tasks. In our case, all of the behaviors involve repositioning objects (i.e., picking up objects and moving them to new locations), which represents a very general class of tasks that might be performed by a robotic arm. While the dataset can be collected in many ways, such as from human demonstrations or prior tasks solved by the robot, we collect it using a set of randomized scripted policies, see Appendix C.2 for details. Since the policies are ran-

---

[1]https://sites.google.com/view/parrot-rl

domized, not every execution of such a policy results in a useful behavior, and we decide to keep or discard a collected trajectory based on a simple predefined rule: if the trajectory collected ends with a successful grasp or rearrangement of any one of the objects in the scene, we add this trajectory to our dataset. We collect a dataset of 50K trajectories, where each trajectory is of length 25 timesteps ($\approx$5-6 seconds), for a total of 1.25m observation-action pairs. The observation is a $48 \times 48$ RGB image, while the actions are continuous 7D vectors. Data collection involves interactions with over 50 everyday objects (see Appendix C.3); the diversity of this dataset enables learning priors that can produce useful behavior when interacting with a new object.

## 5.1 RESULTS, COMPARISONS AND ANALYSIS

To answer the questions posed at the start of this section, we compare PARROT against a number of prior works, as well as ablations of our method. Additional implementation details and hyperparameters can be found in Appendix B.

**Soft-Actor Critic (SAC).** For a basic RL comparison, we compare against the vanilla soft-actor critic algorithm (Haarnoja et al., 2018b), which does not incorporate any previously collected data.

**SAC with demonstrations (BC-SAC).** We compare against a method that incorporates demonstrations to speed up learning of new tasks. In particular, we initialize the SAC policy by performing behavioral cloning on the entire dataset $\mathcal{D}$, and then fine-tune it using SAC. This approach is similar to what has been used in prior work (Rajeswaran et al., 2018), except we use SAC as our RL algorithm. While we did test other methods that are designed to use demonstration data with RL such as DDPGfD (Vecerík et al., 2017) and AWAC (Nair et al., 2020), we found that our simple BC + SAC variant performed better. This somewhat contradicts the results reported in prior work (Nair et al., 2020), but we believe that this is because the prior data is not labeled with rewards (all transitions are assigned a reward of 0), and more powerful demonstration + RL methods require access to these rewards, and subsequently struggle due to the reward misspecification.

**Transfer Learning via Feature Learning (VAE-features).** We compare against prior methods for transfer learning (in RL) that involve learning a robust representation of the input observation. Similar to Higgins et al. (2017b), we train a $\beta$-VAE using the observations in our training set, and train a policy on top of the features learned by this VAE when learning downstream tasks.

**Trajectory modeling and RL (TrajRL).** Ghadirzadeh et al. (2020) model entire trajectories using a VAE, and learn a one-step policy on top of the VAE to solve tasks using RL. Our implementation of this method uses a VAE architecture identical to the original paper's, and we then train a policy using SAC to solve new tasks with the action space induced by the VAE. We performed additional hyperparameter tuning for this comparison, the details of which can be found in Appendix B.

**Hierarchical imitation and RL (HIRL).** Prior works in hierarchical imitation learning (Fox et al., 2017; Shankar & Gupta, 2020) train latent variable models over expert demonstrations to discover options, and later utilize these options to learn long-horizon tasks using RL. While PARROT can also be extended to model the temporal structure in trajectories through conditioning on past states and actions, by modeling $p_{\text{prior}}(a_t, |s_t, s_{t-1}, ..., a_{t-1}, ..., a_0)$ instead of $p_{\text{prior}}(a_t|s_t)$, we focus on a simpler version of the model in this paper that does not condition on the past. In order to provide a fair comparison, we modify the model proposed by Shankar & Gupta (2020) to remove the past conditioning, which then reduces to training a conditional VAE, and performing RL on the action space induced by the latent space of this VAE. This comparison is similar to our proposed approach, but with one crucial difference: the mapping we learn is invertible, and allows the

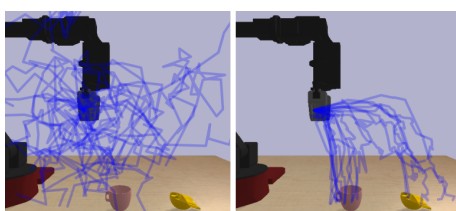

Without Behavioral Prior     With Behavioral Prior

Figure 4: We plot trajectories from executing a random policy, with and without the behavioral prior. We see that the behavioral prior substantially increases the likelihood of executing an action that is likely to lead to a meaningful interaction with an object, while still exploring a diverse set of actions.

RL agent to retain full control over the final actions in the environment (since for every $a \in \mathcal{A}$, there exist some $z = f_\phi^{-1}(a; s)$, while a latent space learned by a VAE provides no such guarantee).

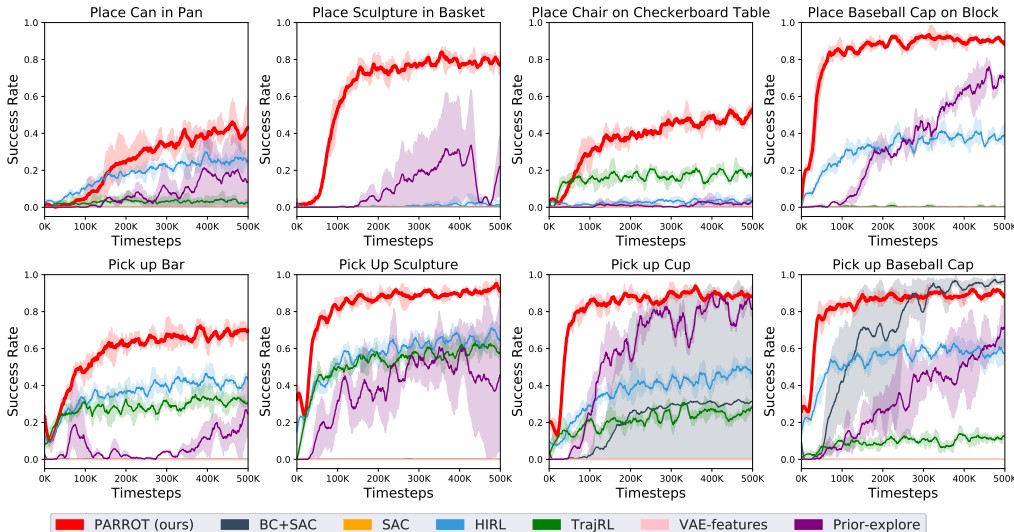

Figure 5: **Results.** The lines represent average performance across multiple random seeds, and the shaded areas represent the standard deviation. PARROT is able to learn much faster than prior methods on a majority of the tasks, and shows little variance across runs (all experiments were run with three random seeds, computational constraints of image-based RL make it difficult to run more seeds). Note that some methods that failed to make any progress on certain tasks (such as "Place Sculpture in Basket") overlap each other with a success rate of zero. SAC and VAE-features fail to make progress on any of the tasks.

**Exploration via behavioral prior (Prior-explore).** We also run experiments with an ablation of our method: instead of using a behavioral prior to transform the MDP being experienced by the RL agent, we use it to simply aid the exploration process. While collecting data, an action is executed from the prior with probability $\epsilon$, else an action is executed from the learned policy. We experimented with $\epsilon = 0.1, 0.3, 0.7, 0.9$, and found $0.9$ to perform best.

**Main results.** Our results are summarised in Figure 5. We see that PARROT is able to solve all of the tasks substantially faster and achieve substantially higher final returns than other methods. The SAC baseline (which does not use any prior data) fails to make progress on any of the tasks, which we suspect is due to the challenge of exploring in sparse reward settings with a randomly initialized policy. Figure 4 illustrates a comparison between using a behavioral prior and a random policy for exploration. The VAE-features baseline similarly fails to make any progress, and due to the same reason: the difficulty of exploration in a sparse reward setting. Initializing the SAC policy with behavior cloning allows it to make progress on only two of the tasks, which is not surprising: a Gaussian policy learned through a behavior cloning loss is not expressive enough to represent the complex, multi-modal action distributions observed in dataset $\mathcal{D}$. Both **TrajRL** and **HIRL** perform much better than any of the other baselines, but their performance plateaus a lot earlier than PAR-ROT. While the initial exploration performance of our learned behavioral prior is not substantially better from these methods (denoted by the initial success rate in the learning curves), the flexibility of the representation it offers (through learning an invertible mapping) allows the RL agent to improve far beyond its initial performance. **Prior-explore**, an ablation of our method, is able to make progress on most tasks, but is unable to learn as fast as our method, and also demonstrates unstable learning on some of the tasks. We suspect this is due to the following reason: while off-policy RL methods like SAC aim to learn from data collected by any policy, they are in practice quite sensitive to the data distribution, and can run into issues if the data collection policy differs substantially from the policy being learned (Kumar et al., 2019).

**Impact of dataset size on performance.** We conducted additional experiments on a subset of our tasks to evaluate how final performance is impacted as a result of dataset size, results from which are shown in Figure 6. As one might expect, the size of the dataset positively correlates with performance, but about 10K trajectories are sufficient for obtaining good performance, and

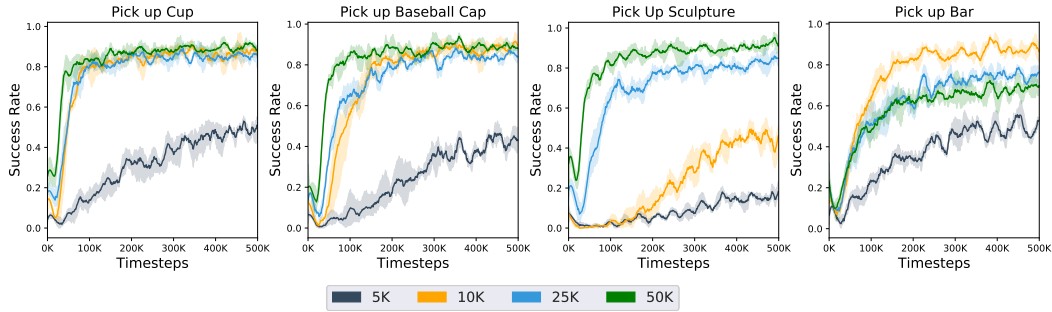

Figure 6: **Impact of dataset size on performance.** We observe that training on 10K, 25K or 50K trajectories yields similar performance.

collecting additional data yields diminishing returns. Note that initializing with even a smaller dataset size (like 5K trajectories) yields much better performance than learning from scratch.

**Mismatch between train and test tasks.** We ran experiments in which we deliberately bias the training dataset so that the training tasks and test tasks are functionally different (i.e. involve substantially different actions), the results from which are shown in Figure 7. We observe that if the prior is trained on pick and place tasks alone, it can still solve downstream grasping tasks well. However, if the prior is trained only on grasping, it is unable to perform well when solving pick and place tasks. We suspect this is due to the fact that pick and place tasks involve a completely new action (that of opening the gripper), which is never observed by the prior if it is trained only on grasping, making it difficult to learn this behavior from scratch for downstream tasks.

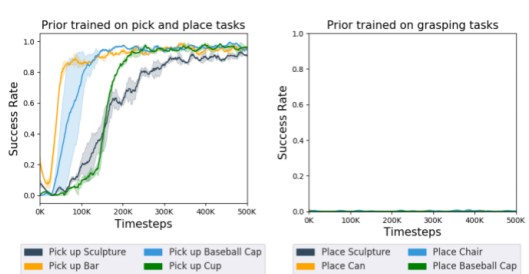

Figure 7: **Impact of train/test mismatch on performance.** Each plot shows results for four tasks. Note that for the pick and place tasks, the performance is close to zero, and the curves mostly overlap each other on the x-axis.

## 6 CONCLUSION

We presented PARROT, a method for learning behavioral priors using successful trials from a wide range of tasks. Learning from priors accelerates RL on new tasks–including manipulating previously unseen objects from high-dimensional image observations–which RL from scratch often fails to learn. Our method also compares favorably to other prior works that use prior data to bootstrap learning for new tasks. While our method learns faster and performs better than prior work in learning novel tasks, it still requires thousands of trials to attain high success rates. Improving this efficiency even further, perhaps inspired by ideas in meta-learning, could be a promising direction for future work. Our work opens the possibility for several exciting future directions. PARROT provides a mapping for executing actions in new environments structurally similar to those of prior tasks. While we primarily utilized this mapping to accelerate learning of new tasks, future work could investigate how it can also enable safe exploration of new environments (Hunt et al., 2020; Rhinehart et al., 2020). While the invertibility of our learned mapping ensures that it is theoretically possible for the RL policy to execute any action in the original MDP, the probability of executing an action can become very low if this action was never seen in the training set. This can be an issue if there is a significant mismatch between the training dataset and the downstream task (as shown in our experiments), and tackling this issue would make for an interesting problem. Since our method speeds up learning using a problem setup that takes into account real world considerations (no rewards for prior data, no need to revisit prior tasks, etc.), we are also excited about its future application to domains like real world robotics.

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

# Appendices

## A  ALGORITHM

**Algorithm 1 RL with Behavioral Priors**

1: **Input**: Dataset $\mathcal{D}$ of state-action pairs $(s, a)$ from previous tasks, new task $M^\star$
2: Learn $f_\phi$ by maximizing the likelihood term in Equation 2
3: **for** step $k$ in $\{1, \ldots, N\}$ **do**
4:     $s \leftarrow$ current observation
5:     Sample $z \sim \pi_\theta(z|s)$
6:     $a \leftarrow f_\phi(z; s)$
7:     $s', r \leftarrow$ Execute $a$ in $M^\star$
8:     Update $\pi_\theta(z|s)$ with $(s, z, s', r)$
9: **end for**
10: **Return**: Policy $\pi_\theta(z|s)$ for task $M^\star$.

## B  IMPLEMENTATION DETAILS AND HYPERPARAMETER TUNING

We now provide details of the neural network architectures and other hyperparameters used in our experiments.

**Behavioral prior.** We use a conditional real NVP with four affine coupling layers as our behavioral prior. The architecture for a single coupling layer is shown in Figure 8. We use a learning rate of $1e-4$ and the Adam (Kingma & Ba, 2015) optimizer to train the behavioral prior for 500K steps.

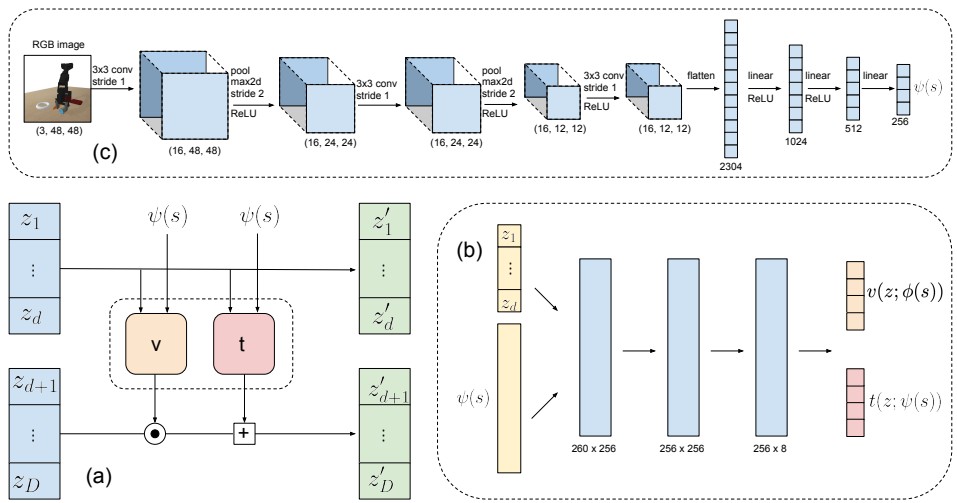

Figure 8: **Coupling layer architecture.** A computation graph for a single affine coupling layer is shown in (a). Given an input noise $z$, the coupling layers transform it into $z'$ through the following operations: $z'_{1:d} = z_{1:d}$ and $z'_{d+1:D} = z_{d+1:D} \odot \exp(v(z_{1:d}; \phi(s))) + t(z_{1:d}; \phi(s))$, where the $v$, $t$ and $\psi$ are functions implemented using neural networks whose architectures are shown in (b) and (c). Since $v$ and $t$ have the same input, they are implemented using a single fully connected neural network (shown in (b)), and the output of this network is split into two. The image encoder, $\psi(s)$ is implemented using a convolutional neural network with parameters shown in (c).

**TrajVAE.** For this comparison, we use the same architecture as Ghadirzadeh et al. (2020). The decoder consists of three fully connected layers with 128, 256, and 512 units respectively. Batch-Norm (Ioffe & Szegedy, 2015) and ReLU nonlinearity are applied after each layer. The encoder is symmetric: 512, 256, and 128 layers, respectively. The size of the latent space is 8 (same as the behavioral prior). We sweep the following values for the $\beta$ parameter (Higgins et al., 2017a): 0.1, 0.01, 0.005, 0.001, 0.0005, and find 0.001 to be optimal. We initialize $\beta$ to zero at the start of training, and anneal it to the target $\beta$ value using a logistic function, achieving half of the target value in

25K steps. We use a learning rate of $1e-4$ and the Adam (Kingma & Ba, 2015) optimizer to train this model for 500K steps.

**HIRL.** This comparison is implemented using a conditional variational autoencoder, and uses an architecture that is similar to the one used by the **TrajVAE**, but with two differences: since this comparison uses image conditioning, we use the same convolutional network $\psi$ as the behavioral prior to encode the image (shown in Figure 8), and pass it as conditioning information to both the encoder and decoder networks. Second, instead of modeling the entire trajectory in a single forward pass, it instead models individual actions, allowing the high-level policy to perform closed-loop control, similar to the behavioral prior model. We sweep the following values for the $\beta$ parameter: 0.1, 0.01, 0.005, 0.001, 0.0005, and find 0.001 to be optimal. We found the annealing process to be essential for obtaining good RL performance using this method. We use a learning rate of $1e-4$ and the Adam (Kingma & Ba, 2015) optimizer to train this model for 500K steps.

**Behavior cloning (BC).** We implement behavior cloning via maximum likelihood with a Gaussian policy (and entropy regularization (Haarnoja et al., 2017)). For both behavior cloning and RL with SAC, we used the same policy network architecture as shown in Figure 9. We train this model for 2M steps, using Adam with a learning rate of $3e^{-4}$.

**VAE-features.** For this comparison, we use the standard VAE architecture used for CIFAR-10 experiments (van den Oord et al., 2017). The encoder consists of two strided convolutional layers (stride 2, window size $4 \times 4$), which is followed by two residual $3 \times 3$ blocks, all of which have 256 hidden units. Each residual block is implemented as ReLU, 3x3 conv, ReLU, 1x1 conv. The decoder is symmetric to the encoder. We train this model for 1.5M steps, using Adam with a learning rate of $1e^{-3}$ and a batch size of 128.

**Soft Actor Critic (SAC).** We use the soft actor critic method (Haarnoja et al., 2018b) as our RL algorithm, with the hyperparameters shown in Table 1. We use the same hyperparameters for all of our RL experiments (our method, HIRL, TrajRL, BC+SAC, SAC).

Table 1: Hyperparameters for soft-actor critic (SAC)

| Hyperparameter | value used |
| --- | --- |
| Target network update period | 1000 steps |
| discount factor $\gamma$ | 0.99 |
| policy learning rate | $3e^{-4}$ |
| Q-function learning rate | $3e^{-4}$ |
| reward scale | 1.0 |
| automatic entropy tuning | enabled |
| number of update steps per env step | 1 |

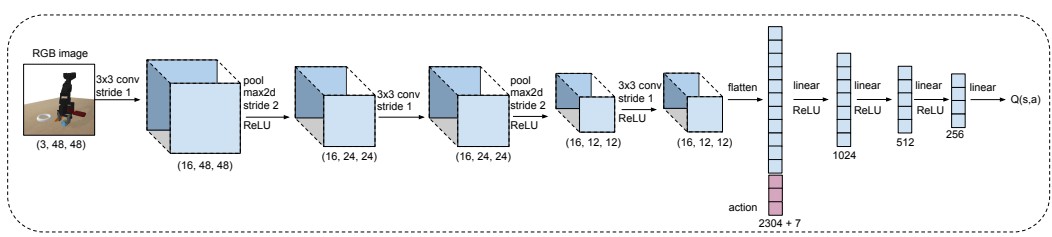

Figure 9: **Policy and Q-function network architectures.** We use a convolutional neural network to represent the Q-function for SAC, shown in this figure. The policy network is identical, except it does not take in an action as an input and outputs a 7D action instead of a scalar Q-value.

## C EXPERIMENTAL SETUP

### C.1 TASKS

We provided a visual depiction of 4 of our 8 evaluations tasks in Figure 3, and the remaining tasks are shown here in Figure 10.

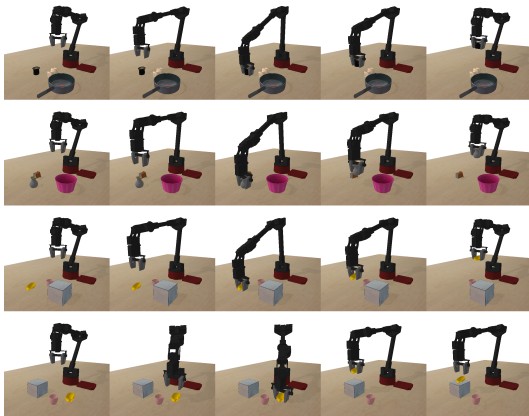

Figure 10: In the first row, the objective is to grasp a can and lift it above a certain height. Rows two and three are similar, except the objective is to grasp a vase and a baseball cap, respectively. The final row depicts a task where the goal is to pick the baseball cap and place it on the marble cube.

## C.2 DATA COLLECTION

We collected our dataset using scripted policies detailed in Algorithms 2 and 3.

---

**Algorithm 2** Scripted Grasping

```
1: threshold ← 0.02
2: numTimesteps ← 25
3: targetPoint ← object position
4: for t in (0, numTimesteps) do
5:     eePos ← end effector position
6:     targetEEDist ← distance(targetPoint, eePos)
7:     if targetEEDist > threshold then
8:         action ← targetPoint − eePos
9:     else if gripperOpened then
10:        action ← close gripper
11:    else if object not raised high enough then
12:        action ← lift upward
13:    else
14:        action ← 0
15:    end if
16:    noise ∼ 𝒩(0, 0.1)
17:    action ← action + noise
18:    s′ ← env.step(action)
19: end for
20:
```

---

**Algorithm 3** Scripted Pick and Place

```
1: threshold ← 0.02
2: numTimesteps ← 25
3: placeAttempted ← False
4: dropPos ← point above container
5: for t in (0, numTimesteps) do
6:     eePos ← end effector position
7:     objectDropDist ← distance(eePos, dropPos)
8:     if placeAttempted then
9:         action ← 0
10:    else if object not grasped AND objectDropDist
       > threshold then
11:        Execute grasp using Algorithm 2
12:    else if objectDropDist > threshold then
13:        action ← dropPos − eePos
14:    else
15:        action ← open gripper
16:        placeAttempted ← True
17:    else
18:        action ← 0
19:    end if
20:    noise ∼ 𝒩(0, 0.1)
21:    action ← action + noise
22:    s′ ← env.step(action)
23: end for
```

---

## C.3 SIMULATION OBJECTS

To collect data in diverse environments, we used 3D object models from the ShapeNet dataset (Chang et al., 2015) and the PyBullet (Coumans & Bai, 2016) object libraries.

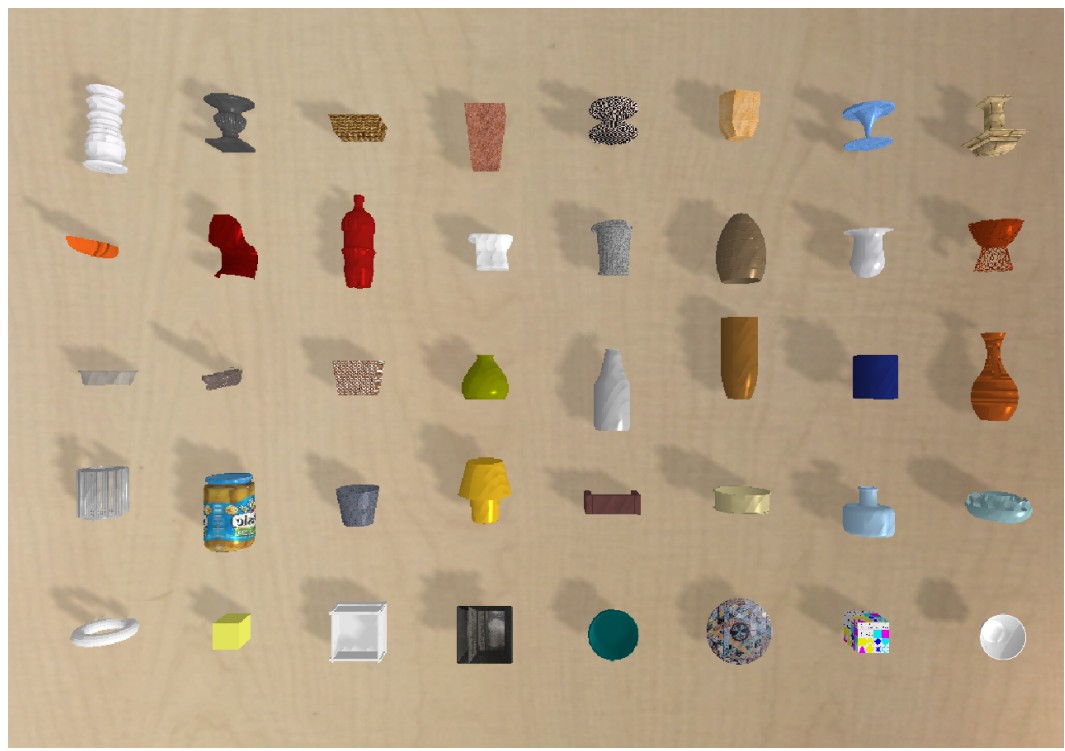

Figure 11: **Train objects**.

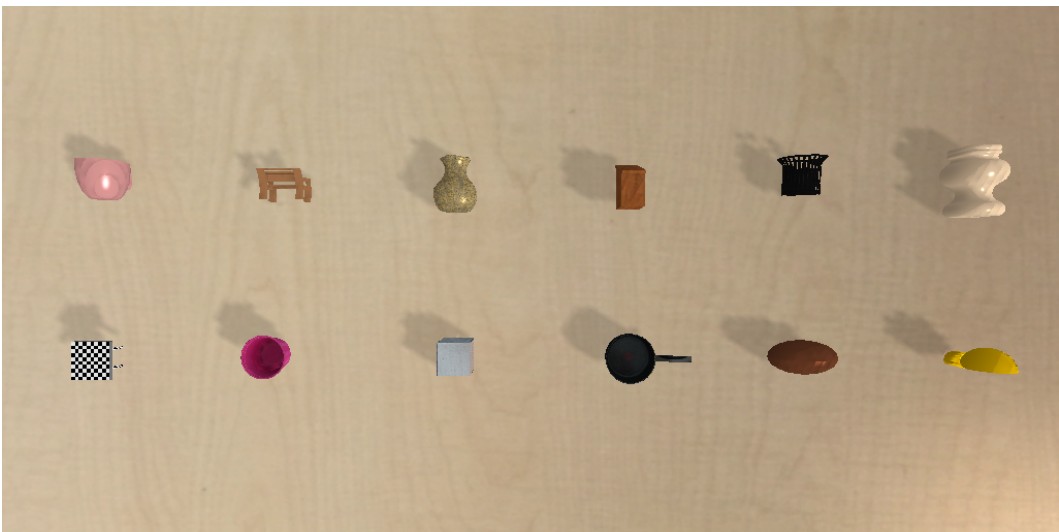

Figure 12: **Test objects**

