# OpenReview forum: "Parrot: Data-Driven Behavioral Priors for Reinforcement Learning"
_ICLR.cc/2021/Conference — ICLR 2021 Oral_

### Official Review · AnonReviewer4 · 2020-10-18
**Very nice work, with some room for additional experiments**

**Rating:** 8
**Confidence:** 3

**Review:**

### Summary

This paper proposes PARROT, a method for learning a policy prior from a dataset of expert state-action pairs that have been derived from multiple similar tasks. The policy prior is parameterized as a deep conditional generative model that maps a noise input and a state to an action. The latter map can be inverted, which is important to guarantee that the prior assigns nonzero probability to the full action space for all states. Given a new task, the policy prior is used to parameterize a new policy; the new policy outputs noise inputs to the policy prior’s invertible mapping, which in turn outputs an action in the original action space. This parameterization of the new policy leads to much more targeted exploration versus sampling actions uniformly from the original action space. Experiments are on a suite of pick-and-place robotic tasks in simulation.

### Pros

- The writing is overall very clear and persuasive. The introduction is especially compelling. The sections are well organized.
- Related work is quite thorough, with just a few potential omissions (see Cons)
- The problem setup section is very much appreciated. It is difficult to formalize or even discuss the assumptions behind an approach like this, but I thought the authors did a very good job.
- The problem setting in general is very motivating. It does seem realistic to suppose that a set of expert trajectories from related tasks are available, but that they are not necessarily annotated with rewards. The authors do a good job explaining how this differs from other problem settings like LfD, meta-learning and meta-imitation learning.
- The whole method is elegant. It is a sensible integration of existing techniques to address a well-motivated problem.
- The experimental results are strong (though there are many opportunities for additional experiments)
- The baselines are well chosen
- The pseudocode in appendix A is clean and clear. All of the notation throughout the paper is too.
- The paper provides a good level of detail in appendix B for reproducibility

### Cons

- Missing references:
    -  “Learning Action Representations for Reinforcement Learning” Chandak et al. ICML 2019
    - “Keep Doing What Worked: Behavioral Modelling Priors for Offline Reinforcement Learning” Siegel et al. ICLR 2020
    - “Residual Policy Learning” Silver et al. (2018) and “Residual Reinforcement Learning for Robot Control” Johannink et al. (2018)
- The empirical results are very encouraging, but I would like to have a better understanding of when the overall approach might fail. Some experiments in this direction would include:
    - Evaluate test-time performance as a function of the number of training tasks
    - Deliberately bias the training tasks so that they non-uniformly sample from the set, and see how much of an effect that has on the downstream performance. For example, train only on pick tasks and test only on pick-and-place tasks, or vice versa.
    - More domains beyond the single one considered
- It would also be nice to see some ablations. In particular, I am curious about the extent to which parameterizing the new task-specific policy with the behavior prior is important, versus just using the behavior prior to gather data. A simple comparison would be to use the behavior prior as an “exploration policy” that is called epsilon% of the time during RL.
- I highly encourage the authors to release code for the paper
- Three random seeds is not enough

### Detailed Comments

- Figure 1 is visually compelling, but a little hard for me to follow. I am most thrown off by the “exploration” part, which says: “attempt all possible tasks in a new scene.” I am interpreting this to be somewhat metaphorical -- it is not as though there is actually a list of tasks, and you are attempting each one in a new scene. Also, as far as I understand, there is not a strict separation between exploration and “learn a specific task via RL” -- exploration is part of that RL process. Let me know if my understanding here is correct. If so, I would recommend perhaps cutting the exploration part of this figure.
- Figure 2 is very helpful and clear
- In the problem setup, there is discussion of fixed state and action space dimensionalities, but it is not stated that the spaces are vector spaces. Would it suffice to instead say that $\mathcal{S}$ and $\mathcal{A}$ are assumed fixed?
- In the problem setup, in two places, there is an expectation over single step rewards that should really be an expectation over temporally discounted returns.
- In the problem setup, it says: “Note, however, that the optimality assumption is not very restrictive: every trajectory can be interpreted as an optimal trajectory for some unknown reward function.” I think this is misleading. It is true that there is some reward function for every trajectory, but the whole point of the problem setup is that there is assumed to be a nontrivial distribution over MDPs, and the important question is whether the observed data are representative of this distribution.
- Please indicate in the caption what the lines and shaded areas represent in Figure 5. I assume they are means and standard deviations, but other choices are plausible too.
- Typos:
    - “Each task can considered a Markov decision processes” (two typos)
    - “(making it expressive)” is missing a period
    - “each coupling layer in the real NVP take as input” (should be “takes”)
    - “hyparparameters”
    - “This comparisons”
    - “behavioural” (I know this is an alternative spelling, but the paper should be internally consistent)

### Questions

-  How does the asymptotic performance of PARROT compare to vanilla RL? Is there a point at which RL starts to outperform PARROT in the tasks considered in this work? It looks like this happens in “Pick up Baseball Cap”.
-  What steps would need to be taken to try PARROT in an environment with discrete actions?
- How might PARROT be used in a continual/lifelong learning setting? My concern is that if the behavior prior is changing over time, all of the policies that have been parameterized using the behavior prior would also change, and would potentially need to be relearned. (This question does not necessarily need to be addressed in the submission -- I am just curious.)
-  What would it take to apply PARROT in a domain where it is not possible to write down good scripted policies (like the ones in Appendix C :) )?

---

> ### Author Response · Authors · 2020-11-19
> **Author response**
>
> We thank the reviewer for their comments. We ran the following set of experiments as suggested by the reviewer:
> 1. **Evaluate the performance of the method as a function of dataset size**. We observe that training on 10K, 25K or 50K trajectories yields similar performance, but performance drops significantly when training on just 5K trajectories from training tasks.
> 2. **Deliberately bias the training dataset so that the training tasks and test tasks are functionally different (i.e. involve substantially different actions)**. We observe that if the prior is trained on pick and place tasks alone, it can still solve downstream grasping tasks well. However, if the prior is trained only on grasping, it is unable to perform well when solving pick and place tasks. We suspect this is due to the fact that pick and place tasks involve a completely new action (that of opening the gripper), which is never observed by the prior if it is trained only on grasping, making it difficult to learn this behavior from scratch for downstream tasks.
> 3. **Use the behavioral prior only for exploration, and learn a policy that directly executes its actions in the environment**. This ablation is able to make progress on most tasks,  but is unable to learn as fast as our method, and also demonstrates unstable learning on some of the tasks. We suspect this is due to the following reason: while off-policy RL methods like SAC aim to learn from data collected by any policy, they are in practice quite sensitive to the data distribution, and can run into issues if the data collection policy differs substantially from the policy being learned [3].
>
> We have also updated the paper with the suggested references, fixed typos and the rewards/discounted return error, and we plan to release the code for the paper as soon as we finish the cleanup process. Here are the answers to the other specific questions asked by the reviewer:
>
>  “Also, as far as I understand, there is not a strict separation between exploration and “learn a specific task via RL” -- exploration is part of that RL process. Let me know if my understanding here is correct.” Yes, your understanding is correct.
>
> “Would it suffice to say that S and A are fixed?” Yes, it would suffice to say so. We have updated this in the problem setup.
>
> “How does the asymptotic performance of PARROT compare to vanilla RL?” We ran our vanilla RL experiments for up to 1m timesteps, and we saw that the performance was essentially flat. We believe this is due to the fact that exploration is challenging in our tasks: a +1 reward is given only on task completion, and the frequency with which a +1 reward is observed during exploration is very low for vanilla RL. Note that in the “Pick up Baseball Cap” problem, it’s the BC+SAC baseline that slightly outperforms PARROT towards the end, not vanilla RL. While we cannot claim that PARROT’s asymptotic performance is better (since we can only run experiments for a finite time), PARROT did outperform all methods in 7 out of 8 tasks in terms of final performance.
>
> “What steps would need to be taken to try PARROT in an environment with discrete actions?” Replacing the real NVP model used in the paper with a model that is suitable for discrete data will allow us to extend PARROT to environments with discrete actions. One such model was proposed by Tran et al [1].
>
> “How might PARROT be used in a continual/lifelong learning setting?”
> In a lifelong learning setting in which the prior changes over time, we could utilize policy distillation [2] to train a new neural network that achieves the same performance as the original policy, but removes dependence on the behavioral prior. This distillation process could be performed soon after the original learning process (or in parallel to it).
>
> “What would it take to apply PARROT in a domain where it is not possible to write down good scripted policies?”
> In domains where it is not possible to write down good scripted policies, we would suggest utilizing one of the following two solutions, depending on the problem:
> 1. Collect demonstrations from human experts, assuming such experts are available.
> 2. Solve prior tasks using reinforcement learning (assuming these tasks are amenable to RL from scratch), use the trajectories generated from the learned policy to speed up learning new tasks.
>
> [1] Discrete Flows: Invertible Generative Models of Discrete Data. Dustin Tran, Keyon Vafa, Kumar Krishna Agrawal, Laurent Dinh, Ben Poole. In Arxiv 1905.10347, 2019.
>
> [2] Policy Distillation. Andrei A. Rusu, Sergio Gomez Colmenarejo, Caglar Gulcehre, Guillaume Desjardins, James Kirkpatrick, Razvan Pascanu, Volodymyr Mnih, Koray Kavukcuoglu, Raia Hadsell. In Arxiv: 1511.06295, 2015.
>
> [3] Stabilizing off-policy q-learning via bootstrapping error reduction. Aviral Kumar, Justin Fu, Matthew Soh, George Tucker, and Sergey Levine. In NeurIPS, 2019.

---

> > ### Comment · AnonReviewer4 · 2020-11-20
> > **Thanks for the additional experiments and insights**
> >
> > Thank you to the authors for the additional experiments and insights. Almost all of my concerns were addressed. Two outstanding concerns that are minor but still worth addressing:
> >
> > - In the problem setup, it says: “Note, however, that the optimality assumption is not very restrictive: every trajectory can be interpreted as an optimal trajectory for some unknown reward function.” I think this is misleading. It is true that there is some reward function for every trajectory, but the whole point of the problem setup is that there is assumed to be a nontrivial distribution over MDPs, and the important question is whether the observed data are representative of this distribution.
> > - Figure 1 is still confusing re: exploration. (Thanks for clarifying my understanding in your response.) I do think removing or changing the exploration part of this figure would make it easier for readers to calibrate their expectations as they start to look at the paper.
> >
> > Otherwise I think the paper is in very good shape.

---

> > > ### Author Response · Authors · 2020-11-20
> > > **Thank you for your response; addressing remaining concerns**
> > >
> > > Thank you for your reply!
> > >
> > > We have updated the paper addressing the remaining concerns:
> > >
> > > 1. We agree that the line about optimality is a bit misleading, and we have removed it from the problem setup. An earlier line in the same section better conveys our assumptions about the prior dataset ("In order for the behavioral prior to be able to accelerate the acquisition of new skills, we assume the behavioral prior is trained on data that structurally resembles potential optimal policies for all or part of the new task.").
> > >
> > > 2. We have updated Figure 1 to remove the exploration part, and edited the text in the figure to indicate that exploration and RL happen in tandem, and not independently of each other.

---

### Official Review · AnonReviewer1 · 2020-10-28
**good method for pre-training for RL; well-written paper**

**Rating:** 7
**Confidence:** 3

**Review:**

This work proposes a method, PARROT, to learn data-driven priors for deep reinforcement learning agents. Motivated by the idea of pre-training with existing data of similar tasks, the authors propose to learn state-conditional behavioral priors from a set of similar tasks for reinforcement learning agents, such that a learning agent explores its environment in a meaningful way. Successful trials from past tasks are used as training data to learn a mapping from pixel-level state input to actions. Experiments in simulated robotic manipulator domain demonstrate the benefit of learning behavioral priors, comparing PARROT against algorithms learning from scratch as well as agents pre-trained with behavioral cloning.

Pros:
The problem setting tackled in this paper is important for applying deep reinforcement learning algorithms on real world robotic tasks;
Training the behavioral prior with PARROT is fully offline, which makes it favorable for practical concerns, as existing meta learning algorithms often require online data (interacting with the environment);
The selection of baseline algorithms in this paper are satisfactory, showcasing the benefits of each design choices made with PARROT;
The paper is well-written and easy to follow in general; the related work section clearly identifies the novelty of PARROT comparing with existing literature

Cons:
PARROT only conducted experiments in a simulated robotic environment, possibly due to the fact that the RL step still takes a considerable amount of data to converge; ideas from works in the imitation learning literature such as GTI by Mandlekar et al. can be leveraged for improving the overall data-efficiency of PARROT


Mandlekar, Ajay, et al. "Learning to Generalize Across Long-Horizon Tasks from Human Demonstrations." Proceedings of Robotics: Science and Systems (RSS), July 2020.

---

> ### Author Response · Authors · 2020-11-19
> **Author response**
>
> We thank the reviewer for their comments. We have included in our discussion that further gains in improving data-efficiency would make PARROT more readily applicable to real world problems.

---

### Official Review · AnonReviewer3 · 2020-10-28
**Overall, I think that this is an interesting paper which shows promising results, but that deserves a better discussion about the related work and the result analysis.**

**Rating:** 6
**Confidence:** 4

**Review:**

In the paper "Parrot: Data-Driven Behavioral Priors for Reinforcement Learning", the authors propose a new approach to leverage previously acquired data to learn a new conditional search space (called representation) that accelerate the convergence of RL algorithm.

The paper is clearly written and well structured. The figures provide a nice illustration of the algorithm and the results are appropriately reported in the graphs. I am just annoyed to see so many important technical details, like the exact definition of tasks, the networks and the different parameters relegated to the appendix, while they are essential to understand the work. However, I understand that is sadly now customary in this venue.

The presented results demonstrate the benefits of the proposed approach, which looks to be at the same time simple and yet very effective. That's great.
However, I have two main issues with the paper:
1) This paper is directly related to the research domain of transfer learning or knowledge transfer, but this is completely ignored in the related work section. In particular, concepts like learning transferable features would appear quite relevant to this work and should also be considered in the experimental reason.
2) I was particularly disappointed by the experimental analysis. More precisely, by the 13 lines of the result section that briefly summarise the content of figure 5. In particular, it shows that PARROT outperforms all the other baselines, great, but there is no further comment/discussion/analysis than that. Ideally, a scientific paper should explain (or at least try to) the observed phenomenon, and thus I was expecting some further analysis that would demonstrate the main hypothesis of the paper, for instance via an ablation study.

Overall, I think that this is an interesting paper which shows promising results, while there are a couple of sections in the paper where the same concepts are repeated multiple times and space could be saved and reused to expand the technical description and the result analysis.

---

> ### Author Response · Authors · 2020-11-11
> **Regarding comparison to transfer learning methods**
>
> Thank you for your comments. We have a question about the first issue you raised: is there a specific prior method of learning transferable features that you would like us to compare against? Certain prior works for transfer learning in RL (such as successor features) are not directly applicable to our problem setting, since (a) we do not require the trajectories in the prior dataset to be annotated with rewards and (b) we learn the behavioral prior from data in an offline fashion, without requiring direct access to the prior tasks. We are currently planning to investigate transfer learning approaches that can be applied to this setting, such as training a VAE on the observations in the prior dataset, and then performing reinforcement learning on top of the features learned by the VAE. Please let us know if there  are any other suggestions you have for this comparison.
>
> We will also update the paper with an expanded analysis of the results presented.

---

> ### Author Response · Authors · 2020-11-19
> **Author response**
>
> *Re transferable features*:
> We have updated the paper with an additional baseline, called VAE-Features in Figure 5. As we mentioned in our last comment, this baseline involves training a variational autoencoder (a beta-VAE) on observations in the prior dataset, and performs reinforcement learning on top of the features learned by this dataset. This baseline is similar to what has been proposed in prior work for transfer learning in RL [1]. In our problem setting, this baseline does not perform particularly well. We believe this is due to the fact that while Higgins et al [1] focus on learning visual features that are robust to small changes in observation (i.e. green background vs pink background), an additional challenge in our problem setting is that of exploration (since a +1 reward is given only when the entire task has been successfully completed). VAE-based methods that tackle exploration challenges via modeling the actions have also been discussed in the paper (see HIRL and TrajRL in Figure 5).
>
>
> *Re analysis*:
> We thank the reviewer for noting that the discussion of experimental results could be expanded. In addition to our existing explanation for why our method outperforms competing methods in the introduction of the Experiments section, we have expanded this discussion in the body of the experiments section, accompanied with the additional ablations and analysis suggested by reviewers. In particular, we have run experiments that
>
> 1. Evaluate the performance of the method as we vary the amount of trajectories we have access to from the training tasks. We observe that training on 10K, 25K or 50K trajectories yields similar performance, but performance drops significantly when training on just 5K trajectories from training tasks.
> 2. Deliberately bias the training dataset so that the training tasks and test tasks are functionally different (i.e. involve substantially different actions). We observe that if the prior is trained on pick and place tasks alone, it can still solve downstream grasping tasks well. However, if the prior is training only on grasping, it is unable to perform well when solving pick and place tasks. We suspect this is due to the fact that pick and place tasks involve a completely new action (that of opening the gripper), which is never observed by the prior if it is only trained on grasping, making it difficult to learn this behavior from scratch down the line.
> 3. Use the behavioral prior only for exploration, and learn a policy that directly executes its actions in the environment. This ablation is able to make progress on most tasks,  but is unable to learn as fast as our method, and also demonstrates unstable learning on some of the tasks. We suspect this is due to the following reason: while off-policy RL methods like SAC aim to learn from data collected by any policy, they are in practice quite sensitive to the data distribution, and can run into issues if the data collection policy differs substantially from the policy being learned [2].
>
> [1] DARLA: Improving Zero-Shot Transfer in Reinforcement Learning. Irina Higgins, Arka Pal, Andrei A. Rusu, Loic Matthey, Christopher P Burgess, Alexander Pritzel, Matthew Botvinick, Charles Blundell, Alexander Lerchner. In ICML, 2017.
>
> [2] Stabilizing off-policy q-learning via bootstrapping error reduction. Aviral Kumar, Justin Fu, Matthew Soh, George Tucker, and Sergey Levine. In NeurIPS, 2019

---

> ### Author Response · Authors · 2020-11-23
> **Request for feedback**
>
> Thank you again for your review. We would like to know if our updated paper (with all major changes detailed in our response below) addresses your concerns, and if you have any additional feedback that you would like to provide.

---

### Official Review · AnonReviewer2 · 2020-10-29
**Novel and valuable work**

**Rating:** 9
**Confidence:** 4

**Review:**

This paper introduces PARROT, a novel approach for pretraining a reinforcement learning agent on near-optimal trajectories by learning a behavioral prior. Essentially, the authors learn a word2vec style embedding of actions for a simple virtual single-arm environment. This embedding will naturally place more common examples from its training data towards the center of the gaussian, making sampling them during training time more likely. The authors demonstrate that this approach outperforms existing pretraining methods in this domain.

This paper presents an interesting novel approach and presents strong support for its value. In particular I appreciate that the basic intuition is relatively straightforward and therefore should be relatively easy to test in new domains. In addition, the evaluation results are impressive.

The paper claims in two instances that human examples would be appropriate for the training data. However, this is not confirmed in the current evaluations to my understanding. The paper indicates that the training data should be “near optimal” multiple times but never explicitly defines what is meant by “near optimal”. Is it that actions must be “useful” (also vague)? More clarity on this would be appreciated.

I have some concerns around certain claims or arguments made in the paper (more on that below). However, this paper still ticks all the boxes for me in terms of novelty and value, and so I would argue for its acceptance.

Some questions for the authors:
1. What is meant by near optimal? Can this be more formally defined?
2. The constraint that the action dimensionalities are fixed seems like a large one. I understand that it’s necessary for the invertible requirement, but I could imagine that a similar approach to this could still be helpful in sim2real problems without this constraint. Would the authors agree?
3. The paper domain is still relatively simple compared to real world problem domains, to what extent do you expect this approach to generalize?

Some smaller points/additional feedback:
- The introduction and abstract are a bit vague, and I didn’t grasp the approach intuition until figure 1. I’d appreciate if some of this information could be moved up into paragraph 3 of the introduction.
-I’d replace “large such datasets” with “such large datasets” or just remove “such”
-I recognize that theoretically that the approach should be able to represent every possible environment action, but it would seem to me still possible that good/useful actions could be placed so far from the center of the Gaussian as to make them very unlikely, depending on the training data. Some discussion about how to avoid this/whether it is a concern would be helpful.

---

> ### Author Response · Authors · 2020-11-19
> **Author response**
>
> We thank the reviewer for their comments.
>
> “What is meant by near-optimal?” Formally, given an MDP with an optimal value function $V^\star (s)$, a near optimal policy $\pi$ is one for which $V^{\pi} (s) ≥ V^\star (s) − \epsilon$. This definition is borrowed from Kearns and Singh [1], who show that, under certain conditions, there exists an algorithm A that can discover such a near-optimal policy in computation time that is polynomial in $\frac{1}{\epsilon}$. We have updated the paper with this reference. In practice, we could obtain such trajectories from RL agents that were trained on these past tasks, expert demonstrations, or hand-crafted solutions. In this paper, we use the latter in the form of scripted policies.
>
> “The constraint that the action dimensionalities are fixed seems like a large one. I understand that it’s necessary for the invertible requirement, but I could imagine that a similar approach to this could still be helpful in sim2real problems without this constraint.”
> As noted by the reviewer, the fixed action dimension is indeed necessary for invertibility. To the best of our knowledge, sim2real problems typically involve agents that have the same action dimension for both the simulated and real versions [2], so we believe this approach could still be applied. For settings where this is not the case, we might need additional modifications to the approach that are beyond the scope of this submission.
>
> “To what extent do you expect this approach to generalize?” We are optimistic about this approach to generalizing to real world problem domains, due to the following reasons: (1) For prior task trajectories, we don’t require access to reward functions. Since rewards can be difficult to obtain in the real world. (2) We demonstrate the effectiveness of our method in problems with high-dimensional observations like images. Such high-dimensional observations are often another challenge we face in real world problems, where we often don’t have direct access to the underlying state of the system. In our opinion, the primary challenges for real world data collection would revolve around collecting large training datasets. In additional experiments we conducted for this rebuttal, we discovered that while 10K trajectories were sufficient to train an effective prior, performance did drop substantially when the number of trajectories was lowered to 5K.
>
> “It would seem to me still possible that good/useful actions could be placed so far from the center of the Gaussian as to make them very unlikely, depending on the training data”
> We agree that this can be a concern, especially if there is a significant mismatch between the training dataset and downstream tasks. While we do not have a concrete answer to how we can avoid this (in this submission), we believe this will make for an interesting direction for future research. We have updated the discussion to reflect this.
>
> [1] Near-Optimal Reinforcement Learning in Polynomial Time. Michael Kearns and Satinder Singh. In Machine Learning, 49, 209–232, 2002.
>
> [2] Sim-to-Real Transfer of Robotic Control with Dynamics Randomization. Xue Bin Peng, Marcin Andrychowicz, Wojciech Zaremba, Pieter Abbeel. In ICRA, 2018.

---

### Author Response · Authors · 2020-11-19
**Author response**

We thank the reviewers for their comments. Based on the feedback, we have expanded upon the experiments and analysis in the paper. All major updates have been highlighted using blue text in the paper PDF.

We have also responded to each of the reviewers individually in the comments below.

---

### Decision · Program_Chairs · 2021-01-07
**Final Decision**

**Decision:**

Accept (Oral)

**Comment:**

This paper presents an elegant and effective approach to knowledge transfer in RL by learning a policy prior from expert data. The paper is generally well structured and well written.
Generally, all the reviewers were favourable about this paper, with its simple idea and convincing results.
It was thought that the paper would benefit from the addition of more discussion around related work, and more experimental results, but it remains a strong paper.